# Clinical Implications of mTOR Expression in Papillary Thyroid Cancer—A Systematic Review

**DOI:** 10.3390/cancers15061665

**Published:** 2023-03-08

**Authors:** Aleksandra Derwich, Monika Sykutera, Barbara Bromińska, Mirosław Andrusiewicz, Marek Ruchała, Nadia Sawicka-Gutaj

**Affiliations:** 1Department of Endocrinology, Metabolism and Internal Medicine, Poznan University of Medical Sciences, 60-355 Poznan, Poland; 2Department of Cell Biology, Poznan University of Medical Sciences, 60-806 Poznan, Poland

**Keywords:** mTOR, mTOR pathway, papillary thyroid cancer, PTC, mTOR inhibitors

## Abstract

**Simple Summary:**

Papillary thyroid cancer (PTC) comprises approximately 80% of all thyroid malignancies. The field of PTC genetics and cancerogenesis remains undetermined. Activated *mTOR* is involved in the development and progression of PTC. We performed a systematic review of papers studying the expression of the *mTOR* gene and its relationship with PTC risk and clinical outcome. We also analyzed the data on mTOR protein expression in PTC and reviewed available data on new targeted therapies and the use of mTOR inhibitors in PTC. A systematic literature search was performed using PubMed, Embase, and Scopus databases (the search date was 2012–2022). Studies investigating the expression of *mTOR* in the peripheral blood or tissue of patients with PTC were deemed eligible for inclusion. Seven of the 286 screened studies met the inclusion criteria for mTOR gene expression and four for mTOR protein expression.

**Abstract:**

Papillary thyroid cancer (PTC) comprises approximately 80% of all thyroid malignancies. Although several etiological factors, such as age, gender, and irradiation, are already known to be involved in the development of PTC, the genetics of cancerogenesis remain undetermined. The mTOR pathway regulates several cellular processes that are critical for tumorigenesis. Activated *mTOR* is involved in the development and progression of PTC. Therefore, we performed a systematic review of papers studying the expression of the *mTOR* gene and protein and its relationship with PTC risk and clinical outcome. A systematic literature search was performed using PubMed, Embase, and Scopus databases (the search date was 2012–2022). Studies investigating the expression of *mTOR* in the peripheral blood or tissue of patients with PTC were deemed eligible for inclusion. Seven of the 286 screened studies met the inclusion criteria for mTOR gene expression and four for mTOR protein expression. We also analyzed the data on mTOR protein expression in PTC. We analyzed the association of *mTOR* expression with papillary thyroid cancer clinicopathological features, such as the TNM stage, BRAF V600E mutation, sex distribution, lymph node and distant metastases, and survival prognosis. Understanding specific factors involved in PTC tumorigenesis provides opportunities for targeted therapies. We also reviewed the possible new targeted therapies and the use of mTOR inhibitors in PTC. This topic requires further research with novel techniques to translate the achieved results to clinical application.

## 1. Introduction

Thyroid cancer is the most common endocrine malignancy [1]. Thyroid cancer incidence has risen in many developed countries over the last decades, but recently, societal guidelines have recommended a less aggressive evaluation and more conservative diagnostic approach, which has resulted in a decline in its incidence [2]. Papillary thyroid cancer (PTC) comprises approximately 80% of all thyroid malignancies and is generally associated with a good prognosis and survival rate when diagnosed early. It can occur at any age, but it is usually detected in the third to fifth decades of the patient’s life, with a mean age of 40. Although several etiological factors, such as age, gender, and irradiation, are already known to be involved in the development of PTC, the genetics of cancerogenesis remain undetermined.

Molecular profiling distinguishes major PTC classes of PTCs characterized by BRAF-predominant, RAS-predominant molecular mutations and RET rearrangements, found in nearly 70% of PTC cases [1,3]. The mTOR pathway is one of the most critical regulatory pathways crucial for tumorigenesis. It is a central cell growth and proliferation regulator in response to environmental and nutritional conditions [4]. In this systematic review, we aim to summarize the current knowledge of mTOR mutations on the pathogenesis and course of PTC.

## 2. Role of mTOR Pathway in Cancerogenesis

The mTOR (mammalian target of rapamycin) protein, also known as FRAP (FKBP12-rapamcyin-associated protein), is a 289 kDa serine/threonine kinase and a member of the phosphatidylinositol 3-kinase-related kinase family of protein kinases. It links growth factors, nutrients, and energy available for cell survival, growth, motility, and proliferation [4]. It is encoded by the MTOR gene [5,6], located on chromosome 1p36.2, is approximately 156 kb in length, and is composed of 59 exons [7]. It has 3434 genetic polymorphisms [8].

The mTOR kinase integrates multiple cell signaling pathways crucial for maintaining cellular homeostasis, including the insulin and growth factor pathway such as for insulin-like growth factors 1 and 2, and the mitogen pathway. The mTOR kinase also monitors cellular energy compounds, ATP levels, and redox status [9]. Signaling by mTOR is commonly activated in tumors and controls cancer cell metabolism by altering key metabolic enzymes’ expression and activity. Numerous studies have demonstrated that the abnormal activation of the mTOR pathway through the stimulation of oncogenes or loss of tumor suppressors contributes to tumor growth, angiogenesis, and metastasis [4,10,11,12,13].

The mTOR kinase comprises two functionally distinct protein complexes: mTOR complex 1 and mTOR complex 2. These mTORC1 and mTORC2 complexes regulate cell growth and metabolism through the direct phosphorylation of critical metabolic enzymes or indirectly through downstream signaling effectors [12]. 

The mTORC1 consists of mTOR, mLST8 (mammalian lethal with SEC13 protein 8/G protein β-subunit-like protein GβL), RAPTOR (regulatory-associated protein of mTOR), and two non-core components: PRAS40 (proline-rich AKT1 substrate 1) and DEPTOR (DEP domain-containing mTOR-interacting protein) [4,14,15,16,17]. The mTORC1 is activated by the PI3K/AKT pathway and inhibited by the TSC1/TSC2 complex [10]. Activation of mTORC1 depends on nutrients and growth factors, including insulin, growth factors, plasma factors, phosphatidic acid, amino acids, and oxidative stress. Leucine, arginine, and glutamine are the most effective activators of mTORC1 among the amino acids [11].

In response to nutrients, mTORC1 is translocated from the cytoplasm to the lysosomal surface, where growth factors activate it through PI3K-AKT signaling. AKT inhibits the TSC1-TSC25 complex, a GTP-activating protein (GAP) for the small GTPase RHEB6. GTP-bound RHEB directly binds and activates mTORC1 in the lysosome [11].

The mTORC1 is inhibited by low levels of energy compounds in the cell, low levels of growth factors, the low redox potential of the cell, caffeine, rapamycin, farnesylthiosalicylic acid (FTS), and curcumin [18,19]. The best-characterized substrates of the mTORC1 complex are p70-S6 kinase (S6K1) and eukaryotic translation initiation factor 4E binding protein 1 (eIF4E binding protein 1, 4E-BP1) [9]. The mTORC1 phosphorylates S6K1 on at least two amino acid residues, stimulating the phosphorylation of S6K1 protein by PDK1 kinase. Activated S6K1 kinase can then promote the initiation of protein synthesis by phosphorylating ribosomal protein S6 and other proteins involved in mRNA translation. S6K1 can also participate in a positive feedback loop by phosphorylating the mTOR kinase molecule.

The mTORC1 complex inhibits the eIF4E binding protein to enhance translation, including the translation of metabolic enzymes and metabolism-related transcription factors. In addition, mTORC1 and S6K directly regulate metabolic enzymes [4,10,11,12,20,21]. 

The mTORC2 includes mTOR, Rictor (rapamycin-insensitive companion of mTOR), mLST8, mSin1 (mammalian stress-activated protein kinase-interacting protein 1), Protor (protein observed with rictor/PRR5, proline-rich protein 5), and DEPTOR [10]. The mTORC2 is an essential regulator of cell cytoskeletal function through interactions with the proteins F-actin, paxillin, RhoA, Rac1, Cdc42, and protein kinase Cα (PKCα) [22]. The mTORC2 promotes metabolism mainly through the activation of AKT kinase. S6K and AKT regulate metabolic enzymes and activate key metabolic transcription factors such as MYC, hypoxia-inducible factor 1α (HIF1α) and HIF2α, FOXO transcription factors, and sterol regulatory element binding protein 1 (SREBP1). The mTORC2 is most likely regulated by insulin, growth factors, plasma factors, and nutrient compound levels [23]. It phosphorylates PKC-α, AKT, and paxillin and regulates the activity of the small GTPases Rac and Rho, associated with the cell survival, migration, and regulation of the actin cytoskeleton [4,22]. In contrast to mTORC1, growth factor signaling alone is sufficient to activate mTORC2, but its mechanism is still incompletely understood [24]. 

The mTOR plays a significant role in diverse types of cancer, including hematologic malignancies and prostate, breast, skin, and head and neck cancers [10,25,26,27,28,29,30,31]. Activation of the mTOR pathway is associated with deregulated production of malignant lymphoid cells and chemotherapeutic resistance in acute lymphoblastic leukemia (ALL). The influence of mTOR has also been discussed in chronic myeloid leukemia (CML) and acute myeloid leukemia (AML). Treatment with dual PI3K/mTOR inhibitors or mTOR kinase inhibitors alone or in combination with conventional ALL therapies or targeted drugs for different cellular cascades can block distinct mechanisms of cell survival in ALL and inhibit cell proliferation and induces apoptosis [25,29,32]. PI3K/AKT/mTOR signaling is active in head and neck cancers, over 90% of which are squamous cell carcinomas [26]. Genetic alterations or ultraviolet (UV) exposure results in the dysregulation of the PI3K/Akt/mTOR pathway in melanocytes, basal cells, squamous cells, or Merkel cells, which leads to the development of melanoma, basal cell carcinoma, cutaneous squamous cell carcinoma, or Merkel cell carcinoma [33]. Clinical studies have already evaluated the use of the PI3K/AKT/mTOR pathway inhibitors in breast, skin, and prostate cancer [27,28,31].

## 3. PTC Cancerogenesis

In the updated 5th edition of the WHO Classification of Thyroid Neoplasm, thyroid tumors are classified based on their cell of origin and pathological and molecular features [34]. Follicular cell-derived tumors are the majority of thyroid neoplasms categorized into benign tumors, low-risk neoplasms, and malignant neoplasms. As we have already mentioned, PTCs are the most common type of thyroid carcinomas and this group comprises many morphological subtypes. Thyroid neoplasms are classified as molecular groups (*BRAF*-like, *RAS*-like, or non-*BRAF*/non-*RAS*-like) based on their mutations and gene expression profiles [35]. Over the recent years, based on studies using the next-generation sequencing (NGS) platform, there has been huge progress in identifying genetic alterations such as point mutations and translocations that play a role in thyroid oncogenesis. Many occur in genes responsible for signaling pathways, particularly the mitogen-activated protein kinase pathway (MAPK), the phosphatidylinositol-3 kinase pathway (PI3K/AKT), and the TSHR cAMP signaling pathway. Understanding the molecular mechanisms underlying thyroid cancers is vital because this knowledge may be useful for prognoses and new therapeutic targets. The report from The Cancer Genome Atlas in 2014 showed that 97% of PTCs have unique molecular alterations (74% with single nucleotide variants, 15% with fusions, 7% with arm-level copy number alterations, and 1% with deletions) [36]. The most common molecular alterations in papillary thyroid carcinomas are *BRAF* (62%)—predominantly, *BRAF*^V600E^—*RAS* (13%), *RET-PTC* (7%), and *TERT* promoter mutation (9%). The other molecular alterations that are less frequent in papillary thyroid carcinomas are *E1F1AX, ALK* fusion, *NTRK1,* or *NTRK3* fusion. Additionally, mutations encoding components of the PI3K-AKT-mTOR pathway, SWI/SNF nucleosome remodeling complex, mismatch repair genes, and histone methyltransferase are exceedingly rare in PTC [37]. The central role in the oncogenesis of thyroid carcinoma is the constitutive activation of the MAPK signaling pathway through mutations or the fusion of its essential proteins such as receptor tyrosine kinases (RTKs, which includes RET, ALK, VEGFR, and TRK), RAS, RAF, MEK, and ERK. The RAS genes (HRAS, KRAS, and NRAS) encode a G-protein (p21) vital in signal transmission from cell membrane receptors, to growth factors, to the nucleus, to the MAPK and PI3K/AKT pathways. Around 10–20% of papillary thyroid carcinomas contain RAS mutations, especially the follicular variant. Point mutations in the NRAS gene are the most frequent among the RAS genes [38]. RET is a proto-oncogene gene that encodes a membrane tyrosine kinase receptor. RET can be activated by fusion with various partners, resulting in the expression of the rearranged RET that activates ret kinase. These rearrangements can also be found in papillary thyroid cancers [39]. The BRAF gene encodes the B-type Raf kinase and is the most potent activator of the MAPK kinase pathway. A point mutation in BRAF is the most common genetic alteration in papillary thyroid cancers. The impact of BRAF mutations on the clinical outcomes of PTC remains debatable. TRK rearrangements are found in only 1–5% of papillary thyroid carcinomas, and at higher frequencies in patients with a history of radiation exposure, the anaplastic lymphoma kinase gene (ALK) was found in 1% of papillary thyroid cancers [40]. The PI3K/AKT/mTOR pathway is an intracellular signaling pathway that regulates major cellular functions, including growth, proliferation, protein synthesis, and autophagy. The mammalian target of rapamycin (mTOR) is a serine/threonine kinase, a downstream effector of the PI3K/AKT pathway, that forms two multiprotein complexes: mTORC1 and mTORC2. The mTORC1 complex affects the transcription of 4EBP1 and p70S6K1, which are involved in mRNA translation. Growth factors, amino acids, ATP, O2 levels, and signaling pathways such as PI3K, MAPK, and AMPK regulate the mTOR signaling pathway. The mTORC2 complex activates PKC-α and AKT and regulates the actin cytoskeleton [4]. Deregulation of mTOR activity (*PI3K* amplification/mutation, *PTEN* loss of function, AKT overexpression, and S6K1, 4EBP1, and eIF4E overexpression) is observed in many types of neoplasms in humans, particularly in melanoma, and has significant effects on the progression of the disease [4,41]. High expression of eIF4E has been observed in medullary thyroid carcinomas (MTCs) as well as in aggressive variants of papillary thyroid carcinomas (PTCs) as compared with conventional PTC and follicular thyroid carcinomas [42]. Therefore, mTOR inhibition is also a promising therapeutic target in the treatment of papillary thyroid carcinoma alone or in combination with inhibitors of other pathways. 

## 4. Materials and Methods

### 4.1. Search Strategy

This study was performed according to the Preferred Reporting Items for Systematic Reviews and Meta-Analyses (PRISMA) guidelines for systematic reviews [43]. The search strategy included terms relevant to mTOR and papillary thyroid cancer and was conducted in three databases (PubMed, Embase, and Scopus) with a date filter of 2012–2022. The following search algorithm was used: (mTOR OR mTOR expression) AND (papillary thyroid cancer OR PTC). Two independent researchers (AD and NSG) performed the literature search. Figure 1 shows the flow of study selection.

### 4.2. Inclusion and Exclusion Criteria

Original clinical studies published in English investigating the expression of *mTOR* in patients with papillary thyroid cancer were deemed eligible for inclusion. The exclusion criteria were: (a) articles published in languages other than English; (b) narrative or systematic reviews and meta-analyses; (c) animal and in vitro studies; (d) case reports, errata, comments, perspectives, letters to the editor, and editorials that did not provide any primary patient data; (e) published abstracts with no available full text; (f) studies that reported cell line expression in thyroid cancer; and (g) studies that included non-PTC patients or tumors with unclear/undetermined histologies. The publication date filter applied was 2012–2022.

### 4.3. Data Extraction

The extraction of the following data was performed: the first author’s name, year of publication, country in which the study was conducted, number of cases and controls, age, gender, method, expression of mTOR in PTC and healthy control groups, tumor-specific data including tumor size, and TNM stage.

## 5. Results

### 5.1. The Expression of the mTOR Gene in PTC (Table 1)

In most of the included studies, DNA was isolated from tissue samples, apart from in one case from a peripheral blood sample; then, the total RNA was extracted using specialized kits. The expression of mTOR was assessed using the polymerase chain reaction (PCR) method or targeted next-generation sequencing (NGS). The results are presented in Table 1. The study conducted by Maruei-Milan et al. [44] on 131 PTC patients and 144 healthy controls found MTOR rs2295080 polymorphism to be associated with a decreased risk of PTC in dominant and allelic models and with a lower risk of a higher tumor stage in the dominant model. The haplotype analysis of MTOR rs2295080 and rs2536 polymorphisms revealed that the frequency of the GT haplotype in PTC subjects was significantly lower than in the controls. The MTOR rs2536 TC genotype and C allele were more frequent in PTC subjects, but no significant association was found between this variant and PTC. Spirina et al. [45] revealed significantly higher expression of mTOR in PTC with the BRAF-V600E mutation. A significantly higher expression of mTOR was observed in the heterogeneous state of the BRAF-V600E mutation in primary tumors and metastases. The other study by this author [46] showed significantly higher mTOR expression in patients with a follicular variant. In the study of a large cohort of 369 PTC samples by Lee et al. [47], MTOR mutation (L1460P, M2387I, L1163V) was found in 3 PTC samples (0.81%). Song et al. [48] compared samples from 50 patients with BRAF-V600E-advanced mutant PTCs, including PTCs with distant metastases and aggressive variants of PTC. Mutations in genes encoding the PI3K/AKT/mTOR pathway members were detected in 16% of patients, although MTOR gene mutation was found in only 4% of classic PTC (2/19 PTC). The overall survival analysis was poorer for patients with PI3K/AKT/mTOR pathway mutations than for those without. Research by Jin et al. [49] and Murugan et al. [50] did not reveal mTOR mutations in PTC samples. The former researchers performed targeted next-generation sequencing in 36 tissue samples from patients with aggressive variants of PTC. Mutations in the PI3K/AKT/mTOR pathway were present in 11% of samples, respectively, but none was the *mTOR* gene mutation itself. The latter researchers screened 53 DTC samples (41 classical papillary thyroid cancer (CPTC), 7 follicular variant (FVPTC), 1 tall cell variant, 1 Hürthle cell cancer, one columnar cell variant PTC, and 2 PDTC) and searched for mutations in exons that were previously reported to be frequently mutated in other human cancers. Neither of the selected tumors revealed *mTOR* mutations. 

**Table 1 cancers-15-01665-t001:** The mTOR gene expression in papillary thyroid cancer.

**Study**	**Country**	**Clinical Data**	**Method** **and Sample**	**Results**
Maruei-Milan et al. 2020 [44]	Iran	131 PCT; 144 HC Male/Female: PCT: 24 (18.3)/107 (81.7) HC: 107 (81.7)/118 (81.9) Age: PCT: 34.6 ± 11.9 HC: 35.6 ± 11.4 TNM stage: I 77 (58.8), II 14 (10.7), III 13 (9.9), IV 10 (7.6), Unknown 17 (13)	PCR-RFLP peripheral blood	- MTOR rs2295080 TG and GG genotypes in the control group were higher than those in PTC subjects (41.7 vs. 32.1% and 8.3 vs. 5.3%), (*p* = 0.06 and *p* = 0.18). - MTOR rs2295080 was found to be associated with a decreased risk of PTC in dominant and allelic models (OR = 0.6; 95% CI = 0.4–0.97; *p* = 0.04, and OR = 0.7; 95% CI = 0.5–0.98; *p* = 0.04). - The haplotype analysis of MTOR rs2295080 and rs2536 polymorphisms revealed that the frequency of the GT haplotype in PTC subjects was significantly lower than in the controls (23.8 vs. 27.8% *p* = 0.023). - MTOR rs2536 TC genotype and C allele were more frequent in PTC subjects, but no significant association was found between this variant and PTC (*p* = 0.08 and *p* = 0.09). - The mTOR rs2295080 TG genotype could reduce the risk of higher tumor stages/TNM stages (III and IV) (OR = 0.3, 95% CI = 0.1–1; *p* = 0.04). The mTOR rs2295080 polymorphism was associated with a lower risk of a higher tumor stage in the dominant model (OR = 0.3, 95% CI = 0.1–0.9; *p* = 0.04).
Spirina et al. 2020 [45]	Russia	20 PTC Male/Female: no data Age: no data TNM stage: T1-4 N0-2 M0	RT-PCR tissue	- The mTOR in PTC without BRAF-V600E mutation 0.26 (0.01; 1.00) vs. 42.72 (12.42; 361.00) in PTC with BRAF-V600E mutation (*p* < 0.05) = significantly higher expression in PCT with BRAF-V600E mutation. - BRAF gene status did not match in the cancers and metastases (Primary tumor and metastasis have the same status BRAF-V600E vs. Primary tumor and metastasis have heterogenous BRAF-V600E) = significantly higher expression of mTOR in heterogenous status mTOR 0.07 (0.01; 0.46) 2.30 (1.00; 653.00).
Spirina et al. 2020 [46]	Russia	41 PTC; 30 cPTC; 11 FVPTC Male/Female: no data Age: cPTC: 50.0 (36.0; 59.0); FVPTC: 60.0 (59.0; 61.0) TNM stage: T1-4 N0-2 M0	RT-PCR tissue	- The mTOR expression in classical variant PCT 2.0 (0.16; 31.0); follicular variant PCT 16.0 (0.0; 19.0) - The mTOR expression in classical variant PCT 2.0 (0.16; 31.0); follicular variant PCT 16.0 (0.0; 19.0) - Significantly higher m-TOR expression in patients with a follicular variant
Lee et al. 2022 [47]	USA	369 PTC Male/Female: no data Age: no data TNM stage: no data	PCR, WES tissue	- MTOR mutations were found in 3 PTC (0.81) (MTOR mutations: L1460P, M2387I, L1163V).
Song et al. 2021 [48]	South Korea	50 PCT (BRAF V600E); 19 cPCT; 23 TCV PCT; 8 PCT metastases Male/Female: 1 (42)/29 (58) Age: 49.5 (38.0–59.5) TNM stage: I 33 (66.0%), II 12 (24.0%), III 1 (2.0%), IV 4 (8.0%)	qPCR, NGS tissue	- MTOR mutation in 4% of PCT (2/19 cPCT)
Jin et al. 2021 [49]	South Korea	36 PTC; 25 TVC PTC; 11 CCV PTC Male/Female: 9 (25)/ 27 (75) Age: 43.5 (34.8–51.0) TNM stage: Stage I 31 (86.1%) Stage II 4 (11.1%) Stage III 1 (2.8%) Stage IV 0 (0.0%)	qPCR, NGS tissue	- MTOR mutation in 0% PCT
Murugan et al. 2015 [50]	Saudi Arabia	63 TC: 41 cPCT; 7 FVPCT; 1 TCV PCT; 1 CCV PCT Male/Female: no data Age: no data TNM stage: no data	PCR tissue	- MTOR mutation in 0% PCT - Rare synonymous genetic variant resulting in C > G transversion (C663G) in 1 out 63 samples (1.6%) - Frequent synonymous variant resulting in C > T transition (C5333T) in 14 out of 84 samples (16%)

Abbreviations: mammalian target of rapamycin (mTOR), papillary thyroid cancer (PTC), healthy controls (HC), follicular variants of PTC (FVPTC), classic PTC (cPTC), tall cell variant papillary thyroid cancer (TCV PTC), columnar cell variant of papillary thyroid carcinoma (CCV PTC), polymerase chain reaction-restriction fragments length polymorphism (PCR-RFLP), real-time polymerase chain reaction (RT-PCR), quantitative polymerase chain reaction (qPCR), whole exome sequencing (WES), next-generation sequencing (NGS).

### 5.2. The Expression of mTOR Protein in PTC (Table 2)

We also analyzed the expression of mTOR protein in PTC in four included studies. The results are shown in Table 2. The expression was assessed with immunohistochemistry. Faustino et al. [51] studied mTOR expression in follicular variants of PTC (FVPTC), classic PTC (cPTC), cPTC metastases, and healthy controls. The enhanced expression of mTOR and phosphorylated mTOR (pmTOR) Ser2448 was found in FVPTC and cPTC when compared with normal thyroid tissue. Total mTOR expression was significantly higher in cPTC relative to the other carcinoma histotypes, and the levels of phosphorylated mTOR in cPTC were significantly higher than in the FVPTC and cPTC metastases. A significant increase in the expression of mTORC1 downstream targets relative to normal thyroid tissue was observed only in cPTC samples. Tavares et al. [52] analyzed 186 PTCs, 119 cPTC, 47 FVPCT, and 20 other variants and observed that phosphorylated mTOR expression indicated tumor aggressiveness in PTC. Higher phosphorylated mTOR expression was associated with the absence of a tumor capsule, the presence of distant metastases, the persistence of disease, and NRAS mutation. Positive pmTOR expression was an independent risk factor for distant metastases and correlated with a greater number of 131I therapies and a lesser expression of sodium iodine symporter. In a study by Ahmed et al. [53], 504 PTC samples were analyzed. Phosphorylation of mTOR was observed in 81 and 39% of PTC samples, respectively. Activation of both mTOR proteins was significantly more common in the younger age group (≤45 y). Statistically significant co-expression of mTORC2 and mTORC1 activity was seen in 32.5% of the PTC. The expression of p-mTORC1 was associated with activated 4E-BP1 expression and p-mTORC2 with p-AKT (and PIK-3CA). Interestingly, p-mTORC1 expression significantly correlated with early stage (I), although no correlation was observed with gender, histology subtype, and extra thyroidal extension. Duman et al. [54] evaluated mTOR expression in differentiated thyroid carcinoma, 82 (81.2%) of which were PTC. No expression was detected in the normal thyroid tissue. In the PTC samples, the expression rate of mTOR was elevated. There was no significant association between mTOR and tumor size and no statistically significant correlation between mTOR expression and lymph vascular invasion, capsular invasion, or multifocality.

**Table 2 cancers-15-01665-t002:** The mTOR protein content in papillary thyroid cancer.

Study	Country	Clinical Data	Method and Sample	Results
Faustino et al. 2012 [51]	Portugal	22 FVPCT; 60 cPCT (23 (38.3%) with BRAF V600E mutation); 21 cPCT metastases; 34 HC Male/Female: no data Age: no data TNM stage: no data	IHC, PCR tissue	- Enhanced expression of mTOR and phosphorylated mTOR (pmTOR) Ser2448 was detected in FVPTC and cPTC, when compared with normal thyroid tissue (*p* ≤ 0.0001 to 0.0022). - Total mTOR expression was significantly higher in cPTC, relative to the other carcinoma histotypes (*p* ≤ 0.0001 to 0.0022). - The levels of pmTOR in cPTC were significantly higher than in FVPTC and cPTC metastases (*p* = 0.0108 and *p* = 0.0005, respectively). - Raptor and rictor were overexpressed (*p* ≤ 0.0001 to 0.0007) in TC in comparison to HC. - A significant increase in the expression of mTORC1 downstream targets relative to normal thyroid tissue, was observed only in cPTC samples (*p* = 0.0180 and *p* < 0.0001, respectively). - Higher expression of mTOR (*p* < 0.0001), pmTOR (*p* = 0.0005), and p4EBP1 Thr37/46 (*p* = 0.0011) was found in primary cPTC than in cPTC metastases. - Significantly higher expression of mTOR (*p* < 0.0001), pmTOR Ser2448 (*p* < 0.0001), raptor (*p* = 0.0037), rictor (*p* = 0.0323) in cPTC BRAF V600E was found than in cPTC BRAF V600E WT.
Ahmed et al. 2014 [53]	Saudi Arabia	536 PCT; 73 FVPCT; 412 cPCT; 19 TV PCT Male/Female: 141 (28)/363 (72) Age: ≤45 y 294 (58.3); >45 y 210 (41.7) TNM stage: I 304 (61.5), II 25 (5.1), III 44 (8.9), IV 121 (24.5)	tissue microarray, IHC tissue	- Co-expression of mTORC2 and mTORC1 activity was seen in a 32.5% (164/504) of the PTC studied and this association was statistically significant (*p* = 0.0244). - The p-mTORC1 expression showed a significant association with the early stage (*p* = 0.0286). - High expression 81.7%; low expression 18.3%
Tavares et al. 2016 [52]	Portugal	191 PCT; 119 cPTC; 47 FVPCT; 20 Other Male/Female: 35 (18)/155 (82) Age: < 45 y 99 (53), ≥ 45 y 87 (47) TNM stage: I 66 (62), II 6 (6), III 25 (23), IV 10 (9)	PCR, IHC tissue	- Higher pmTOR expression was associated with absence of a tumor capsule (*p* = 0.01), presence of distant metastases (*p* = 0.05), persistence of disease (one-year disease-free status and disease-free status at the end of follow-up) (*p* = 0.05), and NRAS mutation (*p* = 0.04). - Positive pmTOR expression showed to be an independent risk factor for distant metastases (odds ratio = 18.2; 95% confidence interval 2.1–157.9; *p* = 0.01). - Higher pmTOR expression was also correlated with a greater number of 131I therapies (r(102)-0.2, *p* = 0.02), greater cumulative dose of RAI (r(100)-0.3, *p* = 0.01), and a lesser expression of sodium iodine symporter (r(44)-0.3, *p* = 0.03).
Duman et al. 2014 [54]	Turkey	101 DTC; 82 (81.2%) PCT Male/Female: PCT: 16 (19.5)/66 (80.5%) Age: 45.32 ± 12.7 TNM stage: I: 56 (68.3%); II: 19 (23.2%); III: 3 (4.9%); IV: 2 (3.7%)	PCR, WES tissue	- No significant association between mTOR and tumor size (*p* = 0.818) - No statistically significant correlation was found between mTOR expression and lymph vascular invasion, capsular invasion, or multifocality (*p* = 0.392, *p* = 0.65 and *p* = 0.156, respectively).

Abbreviations: mammalian target of rapamycin (mTOR), papillary thyroid cancer (PTC), differentiated thyroid cancer (DTC), healthy controls (HC), follicular variants of PTC (FVPTC), classic PTC (cPTC), tall cell variant papillary thyroid cancer (TCV PTC), polymerase chain reaction (PCR), immunohistochemistry (IHC).

## 6. Treatment of PTCs

Most PTCs demonstrate indolent behaviour with an excellent 10-year survival rate of 90%. The first line of treatment is surgery, which is followed by radioactive iodine treatment (RAI) and a suppressive dose of l-thyroxine in high-risk patients [55]. Cervical lymph nodes are the most common area of metastases [56]. Distant metastases are a rare event. They occur in less than 10% of patients, especially in those with more aggressive histological subtypes. Still, one third of those patients can be cured with RAI. The other 60% will be classified as the RAI refractory (RAI-R) during the course of the disease. However, RAI-R is rare, with an incidence of 4–5 cases per million and a 5-year survival rate of less than 50% [57]. Finally, the subgroup of patients that cannot be treated with standard methods is small, but at the same time, is a target for novel therapeutic agents. Recently, extensive genomic studies on DTC carcinogenesis have been performed. One of the most common and important of all the histological types of thyroid cancers is the PI3K/AKT/mTOR pathway.

Preclinical and clinical studies have demonstrated that mTOR pathway targeting may be beneficial in advanced RAI-R DTCs. Several clinical trials have assessed mTOR inhibitors in DTC and some of the themes are still ongoing [58]. The vital compound is everolimus. Its efficacy has been studied as a single agent and in combination therapy (with low-dose cisplatin or pasireotide). Everolimus has exhibited limited therapeutic effects, even when administered simultaneously with various compounds [59,60,61,62,63]. Other combination therapy regimens, such as sirolimus plus cyclophosphamide [64] or sorafenib plus temsirolimus [65], were tested in clinical trials, showing minimal applicability in DTC patients. Those data are presented in Table 3.

Other attractive therapeutic options are multitarget tyrosine kinase inhibitors (TKIs). As they are non-selective, their activity is directed at several molecular spots simultaneously. A blockade of receptors leads to the ceasing of signal transduction in intracellular pathways responsible for cell survival, proliferation, and growth. Vital among these downstream pathways in DTC is the PI3K/AKT/mTOR pathway and MAPK signaling pathway. Additionally, TKI displays an anti-angiogenic effect due to inhibiting receptors for angiogenic factors [66]. Several TKIs have demonstrated clinical efficacy in advanced RAIR DTCs. Still, many ongoing clinical and preclinical trials with various TKIs are being evaluated in thyroid cancer patients. Food and Drug Administration and European Medicines Agency-approved TKIs for advanced RAIR DTC are presented in Table 4 [67]. Sorafenib was approved for clinical use after a large multicenter phase III study named DECISION. There were 417 patients, with 207 in the sorafenib group and 210 on a placebo. The median PFS was 10.8 months in the first group and 5.8 for the control group (*p* < 0.001). The objective response rate (ORR) was 12% in the sorafenib group vs. 0.5% in the placebo group. The stable disease (SD) rate for more than six months was 42% in the sorafenib group vs. 33% in the placebo group [68]. Lenvatinib was approved after a randomized, placebo-controlled, multicenter SELCET trial. A total of 261 patients qualified for the lenvatinib group and 131 for the placebo group. The median PFS was 18.3 in the first group and 3.6 months in the second. The response rate was 64.8 % in the lenvatinib group vs. 1.5% in the control group [69].

Other interesting therapeutic approaches embracing the PI3K/AKT/mTOR pathway in DTC are presented in Table 5.

**Table 3 cancers-15-01665-t003:** The mTOR inhibitors in clinical trials in differentiated thyroid cancer.

Drug Name	Publication	Drug Mechanism	Studied Group	End Points	Results
Everolimus Phase II Clinical Trial	Lim et al. 2013 [7]	mTOR inhibitor	38 patients with various types of thyroid cancer RAI-R or not appropriate for 131I (24 with DTC)	primary: disease control rate: PR + stable response ≥ 12 weeks secondary: response rates; clinical benefit: PD + durable SD; PFS, OS	disease control rate: 81% objective response: 5% of patients SD 76% PD 17% Durable SD (≥ 24 weeks): 45% of patients clinical benefit: 50% patients median PFS: 47 weeks
Everolimus Phase II Clinical Trial	Schneider et al. 2016 [60]	As above	28 progressive metastatic or locally advanced RAI-R DTC 7 patients ATC	primary: disease control rate: CR + PR + SD ≥ 24 weeks secondary: PFS; OS	SD-65% of patients, 58% ≥ 24 weeks Median PFS: 9 months Median OS 18 months
Everolimus Phase II Clinical Trial	Hanna et al. 2018 [61]	As above	33 patients RAI-R DTC 10 patients MTC 7 patients ATC	primary: PFS	DTC cohort: median PFS 12.9 months 2-year PFS 23.6% 2-year OS 73.5%
Everolimus and low-dose cisplatin Phase I Clinical Trial	Fury et al. 2012 [62]	As above plus low-dose cytotoxic chemotherapy	29 patients with various types of advanced solid tumors 7 with unspecified type of thyroid cancer	radiographic response with RECIST	PR: 3 patients Prolonged SD: 5 patients
Everolimus and pasireotide Phase I Clinical Trial	Bauman et al. 2022 [63]	As above plus somatostatin receptor blockade	42 patients DTC 32 (76.2%); MTC 10 (23.8%)	radiographic response with RECIST Arms: A-everolimus B-pasireotide LAR C-combination	OR: 0 Median PFS: 18.3 (A), 1.8 (B), 8.1 (C) months 1-year PFS rates 49.9% (A), 36.4% (B), 25.0% (C)
Sirolimus and cyclophosphamide Retrospective Study	Manohar et al. 2015 [64]	mTOR inhibitor plus cytotoxic agent	15 patients with DTC on sirolimus plus cyclophosphamide 17 patients with DTC on standard care protocol	PFS	1-year PFS rates: 0.45 sirolimus + cyclophosphamide vs. 0.30 control group HR for PFS from initiation of treatment: 1.47
Temsirolimus and sorafenib Phase II Clinical Trial	Sherman et al. 2017 [65]	mTOR inhibitor plus multitarget tyrosine kinase inhibitors (TKI)	36 patients with metastatic RAI-TC of follicular origin	radiographic response rate	PR 22% patients SD 58% patients PD 3% patients 1-year PFS rates 30.5%
Buparlisib Phase II Clinical Trial	Borson-Chazot et al. 2018 [70]	Pan-Class I PI3K Inhibitor	43 patients with advanced RAI-R: DTC in 25, FTC in 17, Hürthle cell carcinoma in 1	primary: PFS at 6 months secondary: OR, PFS at 12 months, OS at 6 and 12 months	probability of PFS was 41.7% at 6 months, 20.9% at 12 months At 6 months: 25.6% patients-SD 48.8%-PD 6 months OS: 85.9% 12 months OS: 78.7%

Abbreviations: complete response (CR); partial response (PR); stable disease (SD); progressive disease (PD); progression-free survival (PFS); overall survival (OS); radioactive iodine refractory (RAI-R); medullary thyroid cancer (MTC); anaplastic thyroid cancer (ATC), differentiated thyroid cancer (DTC); follicular thyroid carcinoma (FTC); objective response (OR); hazard ratio (HR); mammalian target of rapamycin kinase (mTOR), phosphatidylinositol.

**Table 4 cancers-15-01665-t004:** Tyrosine kinase inhibitors approved by the Food and Drug Administration and European Medicines Agency for advanced radioactive iodine refractory differentiated thyroid cancer.

Drug	Target	Indication
sorafenib	RET, c-KIT, VEGFR 1-3, PDGFR, BRAF	RAI-R DTC
lenvatinib	RET, c-KIT, VEGFR 1-3, PDGFR, FGFR	RAI-R DTC
selpercatinib	RET	RAI-R DTC with RET fusion
pralsetinib	RET	RAI-R DTC with RET fusion
larotrectinib	NTRK	Advanced solid tumors with NTRK gene fusion
entrectinib	NTRK, ALK, ROS	Advanced solid tumors with NTRK gene fusion

Abbreviations: radioactive iodine refractory (RAI-R); differentiated thyroid cancer (DTC); rapidly accelerated fibrosarcoma kinase (BRAF); stem cell factor receptor (c-KIT); fibroblast growth factor receptor (FGFR); neurotrophic tyrosine receptor kinase (NTRK); platelet-derived growth factor receptor (PDGFR); rearranged during transfection receptor (RET); c-ros oncogene 1 (ROS); vascular endothelial growth factor (VEGFR); anaplastic lymphoma receptor tyrosine kinase (ALK).

**Table 5 cancers-15-01665-t005:** Therapeutic approaches embracing the PI3K/AKT/mTOR pathway in differentiated thyroid cancer.

Drug Name	Publication	Drug Mechanism	Material	Results
Torin-2	Ahmed et al. (2014) [19]	second-generation mTOR inhibitor	- Tissue samples (536 patients with DTC) In VITRO - PTC cell line: BCPAP and TPC-1 IN VIVO - On mouse TPC-1 tumor xenografts	Immunohistochemistry mTORC1 expression in 81% samples; mTORC2 expression in 39% co-expression 32.5% IN VITRO - prevention of mTORC1 and mTORC2 activity - inhibition of mTOR activity leading to downregulation of cyclin D1 - induction of mitochondrial-mediated apoptosis IN VIVO - decrease in tumor volume and size in mice
CZ415	Li et al. (2018) [71]	mTOR kinase inhibitor blocking mTORC1 and mTORC2 activation simultaneously	- tissue samples (4 PTC patients) In VITRO- TPC-1 human thyroid cancer cell line In VIVO - on mouse TPC-1 cells tumor xenografts thyroid cells	IN VITRO - induction of apoptosis activation - disruption of PTC cell cycle progression - blockade of mTORC1 and mTORC2 activation IN VIVO - significant suppression of tumor growth in mice
paeonol-platinum (II) (PL-Pt[II]) complex	He et al. (2020) [72]	downregulation of mTOR pathway	IN VITRO- ATC cell lines-SW1736 PTC cell lines-BHP7-13 IN VIVO - on mouse SW1736 tumor xenografts	IN VITRO - downregulation of mTOR pathway leading to induction of cytotoxicity - cell apoptosis activation - increase in the sub-G1 cell fraction IN VIVO - reduced tumor volume in mice
OSU-53	Plews et al. (2015) [73]	Dual AMPK activator/mTOR inhibitor	IN VITRO thyroid cancer cell lines of DTC and ATC (BCPAP, TPC1 FTC133, SW1736, and C643)	In VITRO - induction of activation of AMPK - direct inhibition of mTOR activity with consequent suppression of mTOR/p70S6K signaling - autophagy stimulation
NVP-BEZ235	Lin et al. (2012) [74]	dual PI3K/mTOR inhibitor	IN VITRO PTC-BHP7-13, FTC-WRO82-1, undifferentiated FTC-FRO81-2, ATC-8505C, 8305C, KAT4C, KAT18, and MTC-TT human thyroid cancer cell line IN VIVO - on mouse 8505C tumor xenografts	IN VITRO - inhibition of proliferation of all cancer lines - inactivation of signaling downstream of mTORC1 - induction of cell cycle arrest at G0/G1 phase IN VIVO - inhibition of xenografts
Ganetespib	Lin et al. (2017) [75]	heat shock protein 90 inhibitor	IN VITRO - PTC-BHP7-13, FTC-WRO82-1 undifferentiated FTC-FRO81-2, ATC-8505C, 8305C, KAT4C, KAT18, and MTC-TT human thyroid cancer cell lines IN VIVO - On mouse anaplastic and medullary thyroid cancer xenografts	IN VITRO - inhibition of cell proliferation in a dose-dependent manner in all cell lines - induction of arrested cell cycle progression in the G2/M phase - inhibition of expression of proteins involved in the RAS/RAF/ERK and PI3K/AKT/mTOR signaling pathways - induction of apoptosis IN VIVO suppression of tumor growth in mice
(1S,3R)-RSL3 (RSL3)	Sekhar et al. (2022) [76]	small-molecule inhibitor of glutathione peroxidase 4 (GPX4)	In VITRO - PTC cell lines (K1, MDA-T68, MDA-T32, TPC1)	In VITRO - inhibition of mTOR signaling - activation of ferroptosis, which induces cell death and migration - inhibition of DNA response to damage
Canagliflozin	Wang et.al (2022) [77]	glucose cotransporter 2 inhibitor (SGLT2)	- tissue samples (12 PTC and 12 adjacent healthy tissues) IN VITRO PTC cell lines: TPC-1 and BCPAP Nthy-ori-3-1 IN VIVO A tumor xenograft mouse model	Immunohistochemistry Increase in levels of SGLT2 in thyroid cancer in comparison with adjacent tissue In VITRO - inhibition of glucose uptake and glycolysis levels - inhibition of AKT/mTOR activation - induced AMPK activation - Increased apoptosis due to G1/S phase transition arrest IN VIVO:Suppression of tumor growth in mice
PXD101 (Belinostat)	Lin et al. (2013) [78]	histone deacetylase inhibitor	In VITRO - PTC-BHP7-13, FTC-WRO82-1, undifferentiated FTC-FRO81-2, ATC-8505C, 8305C, KAT4C, KAT18, and MTC TT human thyroid cancer cell line In VIVO A tumor xenograft mouse model	IN VITRO - inhibition of cell proliferation accordingly to the dose manner - induction of ROS accumulation - inhibition of the RAS/RAF/ERK and PI3K/mTOR pathways - induction of double-stranded DNA damage and apoptosis IN VIVO - Retardation of tumor growth in mice

Abbreviations: mammalian target of rapamycin kinase (mTOR), phosphatidylinositol 3-kinase (PI3K), papillary thyroid cancer (PTC), medullary thyroid cancer (MTC), anaplastic thyroid cancer (ATC), differentiated thyroid cancer (DTC), follicular thyroid carcinoma (FTC), 5’AMP-activated protein kinase (AMPK), reactive oxygen species (ROS), mechanistic target of rapamycin complex 1 (mTORC1), mechanistic target of rapamycin complex 2 (mTORC2).

## 7. Discussion

The mTOR gene and its signaling pathway are frequently deregulated in human cancer and have become a major therapeutic target. Point mutations of the mTOR gene are expected to play an important role in tumorigenesis and tumoral invasion. Artificially generated mutations of amino acids in mTOR exhibit gain-of-function and oncogenic potential both in vitro and in vivo [79]. The mTOR activation mechanism in human cancers is the epigenetic and genetic alteration of the upstream signaling molecules, methylation of *PTEN*, mutations of *Ras*, *PIK3CA*, and *PTEN*, and the genetic copy gain of receptor tyrosine kinase genes and other genes in the MAP kinase and PI3K/Akt pathways [30,79,80,81,82,83].

### 7.1. Implications of Gene Mutations and Polymorphisms on the Course of PTC

This systematic review summarized the available data on mTOR expression in papillary thyroid carcinoma. Mutations in the mTOR gene in PTC appear to be rare. Mutations in the mTOR gene were found in only 0.81–4% of PTCs [47,48]. Increased mTOR expression is more common in PTC with the BRAF-V600E mutation. Particular polymorphisms are associated with clinical characteristics, including histological types and prognosis. MTOR rs2295080 was less common in PTC, and it was associated with a lower risk of developing cancer and a reduced risk of higher tumor stages. The mTOR rs2536 polymorphism was more common in PTC patients. PI3K/AKT/mTOR alteration increased disease-specific mortality in patients with PTC with the BRAF-V600E mutation, and its effect was independent of the disease stage [84]. The mTOR gene has 3434 genetic polymorphisms [8] and only some of them have been studied in the course of PTC. Although the MTOR rs2536 TC genotype and C allele were more frequent in the PTC subjects, no significant association was found between this variant and PTC compared with normal thyroid tissue. The rs2295080 polymorphism was found to be associated with a decreased risk of PTC in dominant and allelic models [44].

### 7.2. Higher Expression of mTOR Pathway Proteins Might Play a Significant Role in PTC Aggressiveness

The activity of the mTOR protein seems to play a significant role in tumor aggressiveness and might be a risk factor for metastatic potential [52]. Based on our systematic review, a higher expression of mTOR pathway proteins was detected in PTC compared with normal tissue [52,54] and was more common in the younger age group [53]. Higher p-mTORC1 expression was observed in the early stage [53]. Furthermore, higher p-mTOR expression was an independent risk factor for distant metastasis and was associated with a more significant number of 131-I therapies and a greater cumulative dose of RAI [85]. It can be speculated that increased p-mTORC1 expression might be an adverse prognostic factor for patients with PTC diagnosed at early stages. A further focus of research may also be on the role of the mTOR pathway in resistance to 131-I therapy to assess the potential advantages of using pharmacological mTOR blockers in PTC resistance to RAI therapy.

### 7.3. The mTOR Mutations and Pathway Activation in Other Types of Cancers

The signaling of mTOR is activated in other types of clinically aggressive thyroid cancers [42,86,87,88]. In anaplastic thyroid cancer (ATC), whole exome sequencing showed 9% of somatic point mutations (R164Q and M2327I) in the mTOR gene [89]. The mTOR mutation (F2108L) was diagnosed in an ATC patient who was resistant to everolimus treatment [90]. Overactivation of the PI3K/Akt/mTOR pathway plays a vital role in the pathogenesis of medullary thyroid cancer. Most of the carcinogenic effects of RET mutations appear to be mediated by the activation of the PI3K/Akt/mTOR cascade. The oncogenic potential of the human *mTOR* gene has not been established yet. Mutation of the *mTOR* gene has been found only in some cancers, such as renal cancer, breast cancer, and ALL. In a meta-analysis of the genetic polymorphisms of mTOR and cancer risk [7], the mTOR rs2295080 G allele was associated with a significantly higher risk of acute leukemia in the recessive model and a lower risk of genitourinary cancers in the dominant model. In other types of thyroid cancers, most poorly differentiated thyroid carcinomas, mutations in other genes in the PIK3/AKT pathway, including PIK3CA, PTEN, and AKT, have been described [37,91,92,93].

### 7.4. Novel Therapeutic Strategies

Understanding specific factors in the signaling pathways involved in PTC tumorigenesis provides opportunities for targeted therapies. RAI-unresponsive patients demonstrate an increased rate of disease mortality and recurrence. We presented data from preclinical and clinical studies that have demonstrated the beneficial effects of mTOR pathway targeting in this group of patients. Sherman et al. ran a phase II study testing the association of sorafenib and everolimus in 36 patients with DTC and obtained a 53% overall response rate [65]. An open-label phase II clinical trial evaluated the efficacy of everolimus in RAI-refractory thyroid cancer. Out of the 40 patients with follicular cell-origin thyroid cancer, 35% had PTC and 17% had ATC. A total of 72.5% of patients with follicular cell-origin cancer had stable disease and 5% had a partial response [61]. Another phase II study investigated the efficacy and safety of everolimus in locally advanced or metastatic thyroid cancer. Disease control was observed in 31 (81%) patients and the median PFS was 47 weeks in all patients. Calcitonin, CEA, and thyroglobulin concentrations were ≥ 50% lower than the baseline in five (33%) patients with PTC, respectively [59]. In vitro and in vivo studies showed cell growth inhibition in PTC by CZ415, which is a novel, highly efficient, and specific mTOR kinase inhibitor. CZ415 induced apoptotic activation and cell cycle arrest in human PTC cells [71]. Preclinical and clinical studies have demonstrated significant clinical benefits. Further studies are needed to offer a better therapeutic approach for patients with advanced and metastatic thyroid cancer.

### 7.5. Limitations and Further Perspectives

Several limitations of this systematic review should be noted. Although we collected all published clinical evidence investigating mTOR expression in PTC, the number of publications used for this systematic review was small. All studies included were published in English and there might be publications in other languages that contain relevant results. The other restriction is the heterogenicity of the study group. However, it can be concluded that this topic requires further research with novel techniques to translate the achieved results for clinical application.

## 8. Conclusions

Mutations of the *mTOR* gene are rare in PTC and mTOR activity appears to be associated with an independent type of mutation. The activation of the mTOR pathway seems to play a role in the pathogenesis of PTC, especially including mutated BRAF-V600E. In addition, the development of novel therapeutic strategies targeting the mTOR pathway may open a new chapter in the treatment of aggressive PTC.

## Figures and Tables

**Figure 1 cancers-15-01665-f001:**
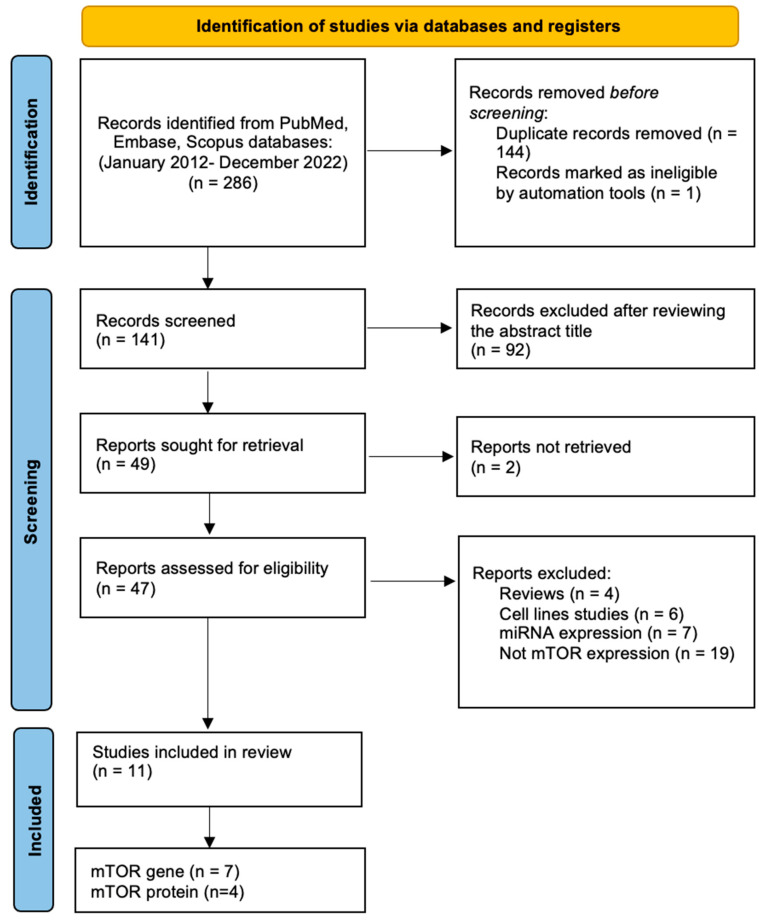
Flow diagram of the study selection.

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
