# Peer review of "Clinical Implications of mTOR Expression in Papillary Thyroid Cancer—A Systematic Review"

_cancers, 2023, doi:10.3390/cancers15061665_

Round 1

Reviewer 1 Report

This is an interesting and very complete review reporting on the expression of mTOR in papillary thyroid cancer and the clinical consequences for the treatment of this disease.

The study is methodologically sound, the results are clearly presented, the discussion is relevant and the paper is well written.

My main comment concerns the Tables 1 and 2 that contain many information and are very difficult to read. I think that for clarity the authors should select only the most important / relevant information or put these tables as supplementary files.

Author Response

Dear Reviewer,

Thank you for taking the time to assess our manuscript. Here, we provide a point-by-point response to your comments.

R1#1

This is an interesting and very complete review reporting on the expression of mTOR in papillary thyroid cancer and the clinical consequences for the treatment of this disease.

The study is methodologically sound, the results are clearly presented, the discussion is relevant and the paper is well written.

My main comment concerns the Tables 1 and 2 that contain many information and are very difficult to read. I think that for clarity the authors should select only the most important / relevant information or put these tables as supplementary files.

Response

We modified the Tables 1 (line 247) and 2 (line 285) by selecting the most relevant information.

R1#2

The first paragraph of the introduction should be rewrote in view of recent data showing a decline in thyroid cancer incidence in the United States. In addition, the bigger incidence in women than in men has been challenged – look for recent publications on the subject.

Response

We rewrote the paragraph about the incidence of thyroid cancer in the introduction (line 42-45). We also changed that information in the abstract (line 21).

Thank you for the opportunity to improve our paper.

Kind regards,

Nadia Sawicka- Gutaj,

Aleksandra Derwich

Reviewer 2 Report

The first paragraph of the introduction should be rewrote in view of recent data showing a decline in thyroid cancer incidence in the United States. In addition, the bigger incidence in women than in men has been challenged – look for recent publications on the subject.

The article needs a thorough spelling and language editing revision. For instance, what do the authors mean by the word “considered” in the sentence: “ BRAF 56 V600E mutations are considered with more aggressive clinicopathological behaviours and play a fundamental role in the tumorigenesis of various thyroid tumours”. Another example of the need of language editing is the number of repeated sentences like in the following paragraph: “Metabolism, growth, and cell behaviour are modulated by various circumstances, the presence of nutrients and growth factors, and altered expression of various genes in- volved in cellular physiology. Acquiring knowledge and understanding the mechanisms  of receiving and integrating extracellular signals in cells and triggering a cascade of intra-  cellular signals that affect cell growth and metabolism is essential for developing effective  treatment strategies. One of the regulatory mechanisms is the mTOR signalling pathway. mTOR is a central regulator of cell growth and proliferation in response to environmental 71 and nutritional conditions. It links growth factors, nutrients, and energy available to cell 72 survival, growth, motility, and proliferation[3].”

The introduction is too long and repetitive. It should end with the purpose of the review, which is unclear.

The description of the results is also very long and should be directed towards clearly pointing out the main findings of each of the analyzed articles without repeating their results.

Table 2 is too busy and difficult to visualize and interpret. In fact, all tables need to be redone with lower content for greater clarity.

The first paragraph of the Treatment of PTCs section should be rewritten since is gives the wrong idea that PTCs are aggressive, which is not the case in most of the cases. The authors cite a 2007 publication to state that 10 to 20% of patients are diagnosed with metastases at the time of diagnosis, not specifying that most metastases are in the lymph nodes, nor updating these data, which no longer correspond to the current demographic presentation of PTC cases.

Unnecessary repetitions all over the text should be eliminated. For instance, why do you need to repeat that “Genetic alterations in the mTOR pathway in DTC have been described in the first part of this article. Here we concentrate on promising 349 pharmacological compounds which recently came to light” on lines 348-350 if you entitled the section 6 “Treatment of PTCs “. ?

I suggest the authors try to draw figures in order to illustrate their conclusions.

Author Response

Dear Reviewer, 

Thank you for taking the time to assess our manuscript. Here, we provide a point-by-point response to your comments.

R2#1

The article needs a thorough spelling and language editing revision.

Response

We corrected the spelling and edited the language.

R2#2

For instance, what do the authors mean by the word “considered” in the sentence: “ BRAF 56 V600E mutations are considered with more aggressive clinicopathological behaviours and play a fundamental role in the tumorigenesis of various thyroid tumours”.

Response

We wanted to highlight the association of BRAF V600E mutation with poor clinical outcomes of PTC. To shorten the introduction, we decided to remove that sentence since BRAF V600E mutations are mentioned in “PTC cancerogenesis” section.

R2#3

Another example of the need of language editing is the number of repeated sentences like in the following paragraph: “Metabolism, growth, and cell behaviour are modulated by various circumstances, the presence of nutrients and growth factors, and altered expression of various genes in- volved in cellular physiology. Acquiring knowledge and understanding the mechanisms  of receiving and integrating extracellular signals in cells and triggering a cascade of intra-  cellular signals that affect cell growth and metabolism is essential for developing effective  treatment strategies. One of the regulatory mechanisms is the mTOR signalling pathway. mTOR is a central regulator of cell growth and proliferation in response to environmental 71 and nutritional conditions. It links growth factors, nutrients, and energy available to cell 72 survival, growth, motility, and proliferation[3].”

Response

We edited the whole paragraph (lines 59-72) and removed repeated sentences.

R2#4

The introduction is too long and repetitive. It should end with the purpose of the review, which is unclear.

Response

We shortened the introduction (lines 42-54) to provide the most relevant information and clarified the aim of our review (lines 56-57). 

R2#5

The description of the results is also very long and should be directed towards clearly pointing out the main findings of each of the analyzed articles without repeating their results.

Response

We rewrote the results section to point out clearly the main findings (lines 216-246 and 257-284).

R2#6

Table 2 is too busy and difficult to visualize and interpret. In fact, all tables need to be redone with lower content for greater clarity.

Response

We modified the Tables 1 (line 247) and 2 (line 285) by selecting the most relevant information.

R2#7

The first paragraph of the Treatment of PTCs section should be rewritten since is gives the wrong idea that PTCs are aggressive, which is not the case in most of the cases. The authors cite a 2007 publication to state that 10 to 20% of patients are diagnosed with metastases at the time of diagnosis, not specifying that most metastases are in the lymph nodes, nor updating these data, which no longer correspond to the current demographic presentation of PTC cases.

Response

We rewrote the first paragraph in the treatment section (lines 295-298) and changed citations (reference 57, 58). It has been changed to “Distant metastases are a rare event. They occur in less than 10 % of patients, especially in those with more aggressive histological subtypes. Still, one third of those patients can be cured with RAI. Other 60%, will be classified as RAI refractory (RAI-R), during the course of the disease. However, RAI-R is rare with incidence of 4–5 cases per million population, the 5-year survival rate is less than 50%[58]. Finally, the subgroup of patients, which could not be treated with standard methods is small, but in the same time it is a target for novel therapeutic agents.” 

R2#8

Unnecessary repetitions all over the text should be eliminated. For instance, why do you need to repeat that “Genetic alterations in the mTOR pathway in DTC have been described in the first part of this article. Here we concentrate on promising 349 pharmacological compounds which recently came to light” on lines 348-350 if you entitled the section 6 “Treatment of PTCs “. ? I suggest the authors try to draw figures to illustrate their conclusions.

Response

We removed the unnecessary repetitions all over the text (lines 52-65, 59-72, 113-114, 296-305)

We also attach the Word file. 

Thank you for the opportunity to improve our paper. 

Round 2

Reviewer 2 Report

The authors followed the suggestions and appropriately modified the text.